# *Eimeria maxima* Rhomboid-like Protein 5 Provided Partial Protection against Homologous Challenge in Forms of Recombinant Protein and DNA Plasmid in Chickens

**DOI:** 10.3390/vaccines10010032

**Published:** 2021-12-27

**Authors:** Di Tian, Xiaoqian Liu, Xiangrui Li, Lixin Xu, Ruofeng Yan, Xiaokai Song

**Affiliations:** MOE Joint International Research Laboratory of Animal Health and Food Safety, College of Veterinary Medicine, Nanjing Agricultural University, Nanjing 210095, China; tiandinjau@163.com (D.T.); 2019107065@njau.edu.cn (X.L.); lixiangrui@njau.edu.cn (X.L.); xulixin@njau.edu.cn (L.X.); yanruofeng@njau.edu.cn (R.Y.)

**Keywords:** *Eimeria maxima*, rhomboid-like protein 5, immunogenicity, protective efficacy

## Abstract

*Eimeria maxima* (*E. maxima*) is one of the most prevalent species that causes chicken coccidiosis on chicken farms. During apicomplexan protozoa invasion, rhomboid-like proteins (ROMs) cleave microneme proteins (MICs), allowing the parasites to fully enter the host cells, which suggests that ROMs have the potential to be candidate antigens for the development of subunit or DNA vaccines against coccidiosis. In this study, a recombinant protein of *E. maxima* ROM5 (rEmROM5) was expressed and purified and was used as a subunit vaccine. The eukaryotic expression plasmid of pVAX–EmROM5 was constructed and was used as a DNA vaccine. Chickens who were two weeks old were vaccinated with the rEmROM5 and pVAX–EmROM5 vaccines twice, with a one-week interval separating the vaccination periods. The transcription and expression of pVAX–EmROM5 in the injected sites were detected through reverse transcription PCR (RT-PCR) and Western blot (WB) assays. The cellular and humoral immune responses that were induced by EmROM5 were determined by detecting the proportion of CD4^+^ and CD8^+^ T lymphocytes, the cytokine levels, and the serum antibody levels. Finally, vaccination-challenge trials were conducted to evaluate the protective efficacy of EmROM5 in forms of the recombinant protein (rEmROM5) and in the DNA plasmid (pVAX–EmROM5) separately. The results showed that rEmROM5 was about 53.64 kDa, which was well purified and recognized by the His-Tag Mouse Monoclonal antibody and the chicken serum against *E. maxima* separately. After vaccination, pVAX–EmROM5 was successfully transcribed and expressed in the injected sites of the chickens. Vaccination with rEmROM5 or pVAX–EmROM5 significantly promoted the proportion of CD4^+^/CD3^+^ and CD8^+^/CD3^+^ T lymphocytes, the mRNA levels of the cytokines IFN-γ, IL-2, IL-4, IL-17, TNF SF15, and IL-10, and specific IgG antibody levels compared to the control groups. The immunization also significantly reduced the weight loss, oocyst production, and intestinal lesions that are caused by *E. maxima* infection. The anticoccidial index (ACI)s of the vaccinated groups were beyond 160, showing moderate protection against *E. maxima* infection. In summary, EmROM5 was able to induce a robust immune response and effective protection against *E. maxima* in chickens in the form of both a recombinant protein and DNA plasmid. Hence, EmROM5 could be used as a candidate antigen for DNA vaccines and subunit vaccines against avian coccidiosis.

## 1. Introduction

Chicken coccidiosis, a globally distributed parasitic disease, is caused by the infection of single or multiple *Eimeria* species, which parasitize in the intestinal epithelial cells of chickens. After being infected by *Eimeria*, chickens typically demonstrate clinical symptoms that usually include loss of appetite, slow growth in weight, diarrhea, and death, etc. [1,2]. All of these pathological conditions eventually lead to a greatly reduced feed utilization rate and growth or laying rate. In severe cases, the mortality rate is as high as 80%, which brings huge losses to the world poultry industry. According to a recent survey, the global cost of coccidiosis in 2016 was estimated to be GBP 10.36 billion, including the losses that were incurred in the production process and the costs of prevention and treatment [3]. There are seven species of *Eimeria* that are recognized in the world, and *E. maxima* is one of the most prevalent species all over the world [4,5,6]. In China, the positive rates of *E. maxima* from 171 farms in Anhui province and 50 farms in Shandong province were 54.67% and 68%, respectively [7,8]. In Australia, 125 samples of commercial chicken flocks from different states and territories were tested, and it was found that the above proportion of *E. maxima* was 58% [9]. Samples from 251 farms in the south of Brazil were collected, and fecal examination showed that *E. maxima* was 63.7% [10].

At present, the main methods to control chicken coccidiosis include the application of anticoccidial drugs and vaccination with live vaccines [11,12]. Although these control strategies play important roles in preventing and controlling the outbreak and epidemic of chicken coccidiosis, alternative control measures are urgently needed due to the emergence of multiple problems such as drug-resistant strains, drug residues in poultry products, as well as the safety issues and high production costs of live vaccines [13]. New vaccines, including DNA vaccines and subunit vaccines, have been suggested as effective strategies for controlling coccidiosis due to their low production cost and high levels of safety, etc. The identification of antigens with good immunogenicity and protective efficacy is the prerequisite for the development of new-generation vaccines. To date, couples of *Eimeria* antigens have been tested as vaccine candidates and have showed promising protective efficacies [14]. However, reports on protective antigens from *E. maxima* are limited compared to *E. tenella* and *E. acervulina*.

Rhomboid-like proteins (ROMs), which are conserved intramembrane serine proteases, are involved in multiple biological activities of various organisms [15]. In Apicomplexan protozoa, ROMs were found to cleave various adhesins that were secreted by the parasites that were mediating the contact and recognition with host cells, promoting the adhesion and invasion of protozoa to host cells [16]. Previous studies on protozoa ROMs have confirmed this function. In *Toxoplasma gondii*, ROMs were reported to cleave the adhesins of TgAMA1, TgMIC2, TgMIC6, and TgMIC12. In *Plasmodium falciparum*, PfAMA1, PfRh1, and PfRh4 were shown to be the substrates of ROMs in vivo [15]. As for the *Eimeria*, EtROM3 was reported to be able to cleave EtMIC4 in *E. tenella* [17]. Given the key role of ROMs in the protozoa invasion process, the protective efficacy of ROMs was evaluated and showed promising protection against protozoa parasites in a couple of studies [18,19,20,21].

In our study, the rhomboid-like protein 5 gene of *Eimeria maxima* (EmROM5) was ligated with prokaryotic and eukaryotic expression vectors to produce the EmROM5 recombinant protein and DNA plasmid. Subsequently, the immune responses and protective efficacy that was induced by rEmROM5 and pVAX-EmROM5 were evaluated, respectively. The results demonstrated the importance of EmROM5 in the development of new vaccines against avian coccidiosis.

## 2. Materials and Methods

### 2.1. Plasmids, Parasites and Animals

The pET-32a (+) plasmid and the pVAX1.0 plasmid were purchased from Novagen (Darmstadt, Germany) and Invitrogen (Carlsbad, CA, USA), respectively. *E. coli* competent cells of DH5α and BL21 (DE3) were obtained from Vazyme (Nanjing, China). One-day-old chicks (Hy-line Variety White) were purchased from a commercial hatchery in Nanjing and were raised in wire cages in coccidia-free conditions. Water and feed without anticoccidial drugs were provided throughout the experiment. Thirty-day-old rats (SD) were purchased from the Qinglong Mountain Breeding Farm in Nanjing. *E. maxima* was propagated by passing through the chickens seven days before the experiment, following previous reports [22]. All animal experiments were conducted with the permission of the Committee on Experimental Animal Welfare and Ethics of Nanjing Agricultural University.

### 2.2. Cloning of EmROM5 Gene

Mini glass beads (0.5 mm diameter) were used to break the oocyst wall in *E. maxima*, following previous reports [22]. After that, the total mRNA was extracted with the E.Z.N.A. Total RNA Kit I (OMEGA, Norcross, GA, USA). Next, mRNA was reverse-transcribed into cDNA with the HiScript IIQ RT SuperMix for qPCR (+gDNA wiper) kit (Vazyme, Nanjing, China). Subsequently, PCR was performed to amplify the EmROM5 gene with the cDNA and specific primers (Table 1). Since the full-length of the EmROM5 gene was not expressed in vitro, the non-transmembrane amino acid sequence of EmROM5 (ntmEmROM5, amino acid sequence: 1-321aa) was selected using TMHMM (http://www.cbs.dtu.dk/services/TMHMM-2.0/, accessed on 15 April 2017) online software for prokaryotic vector construction, protein expression, and further study. The specific primers for ntmEmROM5 and EmROM5 are shown in Table 1. The program was as below: 94 °C, 5 min; 35 cycles (94 °C, 30 s; 55 °C, 30 s; 72 °C, 57 s for ntmEmROM5 or 72 °C, 85 s for EmROM5); and 72 °C, 7 min. The bands of the products were observed by means of 1% agarose gel electrophoresis.

### 2.3. Construction of Recombinant Plasmids pET-32a-ntmEmROM5 and pVAX-EmROM5

The PCR products of ntmEmROM5 and EmROM5 were recovered using the Gel Extraction Kit 200 (OMEGA). Next, ntmEmROM5, EmROM5, the pET-32a vector, and the pVAX1.0 vector were digested by the endonuclease of *Eco*R I and *Xho* I in 10 × H Buffer (Takara, Dalian, China) at 37 °C. The digested fragments of ntmEmROM5 and EmROM5 were ligated into the pET-32a and pVAX1.0 vectors to construct pET-32a-ntmEmROM5 and pVAX-EmROM5, respectively. The ligation products were transformed into DH5α cells for cloning purposes. Finally, these two plasmids were confirmed by DNA sequencing and endonuclease digestion. The obtained DNA sequences were compared using the online tool BLAST (https://blast.ncbi.nlm.nih.gov/Blast.cgi), which was developed by the NCBI. The recombinant plasmid of pVAX-EmROM5 and the empty vector of pVAX for the vaccination trials were prepared using FastPure EndoFree Plasmid Maxi Kits (Vazyme, Nanjing, China) following the manufacturer’s instructions.

### 2.4. Preparation of NtmEmROM5 Recombinant Protein (rEmROM5), Chicken Anti-E. maxima Serum and Rat Anti-rEmROM5 Serum

After the transformation of the pET-32a–ntmEmROM5 plasmid into BL21 (DE3), IPTG (1 mM) was used to induce the expression of rEmROM5. The purification of rEmROM5 was performed using the HisTrap ^TM^ FF Column (Cytiva, Marlborough, MA, USA) following the manufacturer’s instructions. Endotoxins were removed from rEmROM5 using the Endotoxin Removal Kit (Genscript, Nanjing, China) following the manufacturer’s instructions. The concentration of the purified rEmROM5 was measured with the Pierce^TM^ BCA Protein Assay Kit (Thermo Scientific, Waltham, MA, USA) following the manufacturer’s instructions. The protein was diluted to 500 μg/mL with PBS buffer and stored at −80 °C. Simultaneously, the pET-32a tag protein was obtained using the same procedure.

To prepare chicken anti-*E. maxima* serum, fourteen-day-old chickens were artificially infected with 1 × 10^4^
*E. maxima* oocysts 5 times at one-week intervals by means of oral administration. The blood samples were collected by cardiac puncture 7 days after the last infection day. To prepare rat anti-rEmROM5 serum, 30-day-old rats were immunized with an emulsion consisting of 0.5 mL (500 μg/mL) rEmROM5 and 0.5 mL of Freund’s complete adjuvant (Sigma-Aldrich, Merck KGaA, Darmstadt, Germany) through subcutaneous injection on the back. A total of 14 days later, the rats were immunized with another emulsion consisting of 0.5 mL rEmROM5 and 0.5 mL Freund’s incomplete adjuvant. After that, the rats were given three more immunization using the same dose and component as the last immunization cycle, and these injections were given at one-week intervals. Finally, blood was collected from the fundus vein, the serum titer was detected by indirect ELISA, and after the titer reached the appropriate level, the rats were killed for blood collection, and the serum was separated and stored at −70 °C.

### 2.5. Western Blot Analysis of rEmROM5

The recombinant protein of EmROM5 was analyzed by Western blot assays. After the SDS-PAGE of rEmROM5, it was transferred to nitrocellulose membranes (Merck Millipore, Tullagreen, Carrigtwohill, Ireland). Then, the membranes were incubated in the primary antibody of chicken anti-*E. maxima* serum (1:50 dilution) or His-Tag Mouse Monoclonal antibody (1:8000 dilution, Proteintech, Wuhan, China)at ambient temperature for 4 h separately, and incubated in the secondary antibody of horseradish peroxidase (HRP)-conjugated goat anti-chicken IgG (1:4000 dilution, Biodragon-immunotech, Beijing, China) or HRP-conjugated anti-mouse (1:10,000 dilution, Thermo Fisher Scientific, Waltham, MA, USA) at 37 °C for 1.5 h separately. Meanwhile, uninfected chicken serum was set for negative control as a primary antibody. Finally, an Enhanced HRP-DAB substrate chromogenic kit (TIANGEN Biotech, Beijing, China) or ECL chemiluminescence detection kit (Vazyme, Nanjing, China) was used for color rendering.

### 2.6. Reverse Transcription PCR and Western Blot Analysis of Transcription and Expression of pVAX-EmROM5 In Vivo

Fourteen-day-old chickens were injected with pVAX–EmROM5 or pVAX1.0 in the leg muscle with a dose of 100 μg per chicken. The injection sites were marked. After 7 days, muscles were collected from the injection sites and non-injection sites. To detect the transcription of EmROM5, 1 g of muscle was removed from the injection sites and was ground with 1 mL RNAiso Plus (TaKaRa) for 30 min in ice water. The total mRNA of the muscles were extracted following the instructions for the RNAiso Plus reagent, and DNase I (TaKaRa) was used to eliminate the residual recombinant plasmid of the muscles. The EmROM5 gene primers (Table 1) were used for RT-PCR with the mRNA products as templates. Electrophoresis was conducted to detect the product bands to reflect the transcription of EmROM5.

To detect the expression of EmROM5, the removed muscles were treated with RIPA Lysis Buffer (Strong, CWBIO, Beijing, China) for 2 h and were centrifuged at 12,000· *g* for 15 min, and the supernatant was used for SDS-PAGE. During the WB analysis to detect the level of protein expression, rat anti-rEmROM5 serum (1:50 dilution) and HRP-conjugated goat anti-rat IgG (1:4000 dilution, Biodragon-immunotech, Beijing, China) were used as the primary antibody and secondary antibody, respectively. Meanwhile, negative rat serum was set for the negative control as a primary antibody.

### 2.7. Determination of Immune Response Induced by EmROM5 in Chickens

#### 2.7.1. Animal Immunization

There were five groups of fourteen-day-old chickens with similar weights. Experimental group chickens were vaccinated with 200 μg of rEmROM5 or 100 μg of pVAX–EmROM5 by injecting into the leg muscles, respectively. Control group chickens were injected with the pET-32a tag protein, pVAX1.0 plasmid, or PBS, respectively. After seven days, a booster immunization was performed using the same procedure.

#### 2.7.2. Determination of EmROM5-Induced Changes in Spleen T Lymphocyte Subpopulations by Flow Cytometry

On the 7th day after each immunization, five chickens in each group were dissected, and their spleens were removed. The spleens were well ground in PBS buffer and were filtered with a 200-mesh cell sieve; the filtrate was slowly added along the wall into a 10 mL sharp-bottomed glass centrifuge tube containing 5 mL of 37 °C pre-warmed lymphocyte separation solution (TBDscience, Tianjin, China) and was then centrifuged at 720 g for 16 min; the middle white layer of the cells was transferred into new centrifuge tubes and were washed twice with PBS buffer. Finally, the lymphocytes were counted using a blood counting chamber, and the density was adjusted to 1 × 10^7^ cell/mL by PBS buffer. An amount of 100 μL of counted lymphocytes were taken from each group and were placed into 2 mL centrifuge tubes; lymphocytes from all of the groups were double stained with mouse anti-chicken CD3-FITC (Southernbiotech, Birmingham, AL, USA) and mouse anti-chicken CD4-PE or mouse anti-chicken CD8-PE by incubating at 4 °C for 45 min under dark conditions; in addition, lymphocytes from the PBS control group were treated with blank or single stain for template adjustment. The sample test was run using a FACScan flow cytometer (BD Biosciences).

#### 2.7.3. Determination of EmROM5-Induced Changes in Cytokines by Quantitative Real-Time PCR

Specific PCR primers of GAPDH, IFN-γ, IL-2, IL-4, IL-17, TNF SF15, TGF-β, and IL-10 cytokines were designed and synthesized (Table 2). The total mRNA of the spleen lymphocytes from blank chickens was extracted and reverse-transcribed into cDNA as a template. A screening experiment with a series of template concentration gradients was conducted to determine the amplification efficiencies of the gene primers [23]. There were 18 reactions per primer pair, the template was diluted to 5 different concentration gradients, each gradient was repeated 3 times, and 3 replicates of no template control (NTC) were set up. The amplification efficiency (E) and average C_q_ (ΔC_q_) of the primers were calculated, and the cytokine primers ranging in amplification efficiency from 90% to 110% and with a ΔC_q_ of 3 or greater were selected. A slope of −3.32 represents 100% PCR efficiency, and the formula is as follows: E = 10^−1/slope^ − 1 [24]; ΔC_q_ = C_q(NTC)_ − C_q(lowest input)_. The primers with good amplification efficiency were used to determine the samples of the experimental groups. The reaction system and reaction procedure refer to the instruction of the ChamQ^TM^ SYBR qPCR Mix (Vazyme, Nanjing, China). The relative quantification of the cytokine mRNA compared to that of the internal reference gene (n-fold change to the PBS buffer control group) was estimated using the 2^−^^ΔΔCt^ method [25].

#### 2.7.4. Determination of EmROM5-Specific IgG Antibody Level by Indirect ELISA

One week after the first and the second immunizations, blood was drawn from the heart of every chicken. The fresh blood was placed in an incubator that was set at 37 °C for 2 h and was then placed in a fridge that was set at 4 °C for 4 h. After centrifuging at 540× *g* for 8 min, the serum was separated and stored at −30 °C. Then, the rEmROM5-specific IgG serum level was determined by indirect ELISA on the basis of the reported method [26]. First, flat-bottomed 96-well plates (MarxiSorp, Nunc, Denmark) were coated with rEmROM5 that had been diluted in the coating buffer (0.05 M carbonate buffer, pH = 9.6). Second, the chicken serum that had been collected in the previous step (1:50 dilution) was used as the primary antibody, and the HRP-conjugated goat anti-chicken IgG (1:4000 dilution) was as the secondary antibody. Third, color production was conducted with 3, 3’, 5, 5’-tetramethylbenzidine (TMB, TIANGEN), and the OD450 was determined using a microplate reader (Thermo Fisher Multiskan FC).

### 2.8. Assessment of Protective Efficacy of EmROM5 against Challenge with E. maxima

Two vaccination-challenge trials were performed to evaluate the protective efficacies of rEmROM5 (Trial 1) and pVAX–EmROM5 (Trial 2) separately; chickens with similar growth status were randomly divided into eight groups (Table 3). The experimental groups had 200 μg of rEmROM5 (without adjuvant) or 100 μg of naked plasmid pVAX–EmROM5 injected into their leg muscles at two weeks and three weeks of age, respectively. The challenged and unchallenged control groups were injected with PBS. The pET-32a tag protein and pVAX1.0 control groups were injected with the same amount of tag protein or empty plasmid as the corresponding experimental groups. At four weeks of age, the two groups of unchallenged chickens were given PBS orally, the other six groups were orally infected with 1 × 10^5^ freshly sporulated *E. maxima* oocysts [27]. All of the chickens were slaughtered six days post challenge infection.

The survival rate, intestinal lesion score, weight gain, and oocysts output were recorded and were used to evaluate the protective efficacy of the vaccines. The survival rate was counted as follows: the amount of surviving chickens/the amount of initial chickens. The enteric lesion score was recorded following the method described by Johnson and Reid (1970) [28]. Oocysts of per gram feces (OPG) were determined via McMaster’s counting technique [29,30]. The anticoccidial index (ACI) is a comprehensive index to evaluate the anticoccidial efficacy of vaccines/drugs and is calculated as follows: (survival rate + relative rate of weight gain) − (lesion value + oocyst value) [2,31].

### 2.9. Statistical Analysis

All data were analyzed using the one-way ANOVA followed by Duncan’s multiple range test using IBM SPSS Statistics 20 software. Since lesion scores and oocyst output do not follow the normal distribution, statistical analyses were carried out with pairwise comparisons using the Wilcoxon rank sum test. Data were presented as the mean ± SD. The statistical significance was set at *p* < 0.05.

## 3. Results

### 3.1. Cloning of NtmEmROM5 and EmROM5, Construction of pET-32a-ntmEmROM5 and pVAX-EmROM5

EmROM5 and ntmEmROM5 were amplified through RT-PCR. As shown in Figure 1, agarose gel electrophoresis showed bands that were approximately 963 bp and 1461 bp in size that were equal to the molecular weights of ntmEmROM5 (Figure 1A, Lane 1) and EmROM5(Figure 1B, Lane 2), respectively. Recombinant plasmids of pET-32a–ntmEmROM5 and pVAX–EmROM5 were constructed and identified by enzyme digestion, two bands with the sizes of approximately 963 bp (Figure 1C, Lane 3) and 1461 bp (Figure 1D, Lane 4) were observed, which were consistent with the sizes of ntmEmROM5 and EmROM5, respectively. Moreover, the sequencing results showed that the nucleotide sequences of the two genes shared 100% similarity with the sequence in GenBank (ID: XM_013478359.1).

### 3.2. Expression and Western Blot Analysis of rEmROM5

The rEmROM5 was expressed in *E. coli* and was analyzed through WB analysis. As shown in Figure 2, the expression of rEmROM5 was positively correlated with the induction time (Figure 2A, Lane 4–8). The expressed rEmROM5 was mainly distributed in the inclusion bodies of the host bacteria (Figure 2B, Lane 11). After purification, SDS-PAGE showed a single protein band close to 53.64 kDa, which is consistent with the total molecular weight of the ntmEmROM5 protein (35.31 kDa) and pET-32a tag protein (18.33 kDa) (Figure 2B, Lane 12). WB analysis showed that rEmROM5 was recognized by His-Tag Mouse Monoclonal antibody (Figure 2C, Lane 3) and chicken serum against *E. maxima* (Figure 2C, Lane 1) separately. Meanwhile, rEmROM5 was not recognized by the negative serum (Figure 2C, Lane 2). The original images for Figure 2C are shown in Appendix A.

### 3.3. Transcription and Expression of pVAX-EmROM5 in the Injection Site Muscles of Chickens

RT-PCR was used to detect the transcription of the EmROM5 gene in the injection site muscles of the chickens. In Figure 3A, a DNA band of approximately 1461 bp was detected from the pVAX–EmROM5-injected muscles (Figure 3A, Lane 1), and there were no bands in the samples of the pVAX l.0-injected muscles and non-injected muscles (Figure 3A, Lanes 2 and 3). WB analysis was used to detect EmROM5 gene expression in the injected muscles. As per the results in Figure 3B, a protein band of 53.57 kDa that is consistent with the molecular weight of the EmROM5 protein was detected from the pVAX–EmROM5-injected muscles (Figure 3B, Lane 1), and no bands are visible for the negative control (Figure 3B, Lane 2). These results indicate that successful pVAX–EmROM5 transcription and expression of occurred in the injected muscles. The original images for Figure 3B are shown in Appendix A and Appendix A.

### 3.4. Changes of CD4+/CD3+ and CD8+/CD3+ T Lymphocyte Subpopulation in the EmROM5 Immunized Chickens

The proportions of the T lymphocyte subpopulation the spleens were detected by flow cytometry to reflect the changes that were induced in the T lymphocytes by EmROM5. The results are shown in Table 4. Compared to the pET-32a tag protein and PBS groups, in the pVAX1.0 and PBS groups, the two proportions of T lymphocytes in the rEmROM5- and pVAX–EmROM5-immunized groups were significantly increased seven days after these two immunizations (*p* < 0.05). There is no difference in the two T lymphocyte proportions between the pET-32a tag protein, pVAX1.0, and PBS groups in the same set of data (*p* > 0.05).

### 3.5. Changes of Cytokines Transcription in Splenic Lymphocytes in the EmROM5 Immunized Chickens

Changes in splenic lymphocyte cytokines that were induced by EmROM5 were detected by qPCR. The relative changes in the mRNA transcription levels of seven cytokines, IFN-γ, IL-2, IL-4, IL-17, TNF SF15, TGF-β, and IL-10, in the splenic lymphocytes are shown in Figure 4. In the pET-32a–EmROM5 recombinant protein (rEmROM5) immunization group, the mRNA levels of all of the above cytokines, except TGF-β, were significantly increased compared to those in the PBS and pET-32a tag protein groups seven days after the first immunization (*p* < 0.05), while the mRNA levels of all seven of the cytokines increased significantly seven days after the second immunization (*p* < 0.05). In the pVAX–EmROM5 immunization group, the mRNA levels of the seven cytokines increased significantly compared to in the PBS and pVAX 1.0 groups seven days after the first and second immunization (*p* < 0.05).

### 3.6. Specific Antibody IgG Levels in Chicken Serum after Immunization

The levels of the specific antibody IgG in the serum from the chickens who had been immunized with rEmROM5 and pVAX–EmROM5 were detected by indirect ELISA (Figure 5). Compared to the three control groups, the levels of specific antibody IgG in the two immunized groups were significantly increased one week after the first and second immunization (*p* < 0.05) period. Furthermore, the antibody level one week after the second immunization was higher than that after the first immunization. No significant differences were observed between the PBS and pET-32a tag protein or pVAX1.0 groups (*p* > 0.05).

### 3.7. Protective Efficacy of EmROM5 against Challenge with E. maxima

Two vaccination challenge trials were performed to assess the protective efficacies of rEmROM5 and pVAX–EmROM5 against the challenge of *E. maxima*. The results are shown in Table 3. Compared to the unchallenged control groups in trial 1 and trial 2, the average weight gains in the four challenged control groups (the pET-32a tag protein and pVAX1.0 control groups were challenged groups in trial 1 and trial 2) were significantly decreased (*p* < 0.05). Compared to the four challenged control groups, immunization with rEmROM5 or pVAX–EmROM5 significantly increased weight gain in the immunized chickens (*p* < 0.05). Moreover, the average OPG and enteric lesions of the rEmROM5- and pVAX–EmROM5-immunized chickens were significantly lower than those of the four challenged control groups (*p* < 0.05); the ACI of the immunized chickens were above 160, which indicated that the recombinant protein and plasmid provided moderate protective efficacy against *E. maxima* infection in chickens.

## 4. Discussion

Coccidiosis is a highly infectious parasitic disease and causes great losses to the world poultry industry [32]. In the current strategies for controlling chicken coccidiosis, anticoccidial drugs have the disadvantages of drug resistance and drug residue, and traditional live vaccines also create safety and cost problems. Hence, genetically engineering vaccines, including subunit vaccines and DNA vaccines, has been considered to be a prospective alternative measure against coccidiosis [14,33,34,35,36]. *E. maxima* is one of the most prevalent species that cause coccidiosis in clinics; however, there are limited reports on new vaccine antigens against *E. maxima*. To find a new vaccine antigen, it is essential to study its immune protection. Several studies have reported evaluations of the immune protective effect of *E. maxima* antigens; for example, EmMIC2 and gametocyte antigen Gam82 of *E. maxima* both have good immune effects against *E. maxima* infection in chickens and can be in the development of new vaccines [37,38]. Our study evaluated the immunogenicity and protective effect of EmROM5 and found that EmROM5 induced significant immune responses and produced moderate protection against *E. maxima* challenge. Our study enriches the candidate antigens for the development of new types of vaccines against *E. maxima.*

In apicomplexan protozoa, invasion-related molecules are always considered to be potential candidate antigens for the development of new vaccines and show promising protection levels against protozoa infection. For example, in *Toxoplasma gondii* and *Plasmodium falciparum*, apical membrane antigen 1(AMA1) binds to its receptor rhoptry neck 2(RON2); their interaction promotes the invasion of protozoa into the host cell [39]. Various animal protection experiments have proven that AMA1 has developed as an effective candidate vaccine antigen against many apicomplexan protozoa [40,41]. In coccidiosis vaccine research, microneme proteins (MICs) are considered to be ideal protective antigens. MICs are secreted by microneme to the surface of the parasite cell membrane, which can recognize and bind with the specific ligands on the host cell surface and can play key roles in the attachment and invasion of the *Eimeria* sporozoite [42]. The protective efficacy of the MICs of various *Eimeria* (EtMIC1, EtMIC2, EaMIC2, EmMIC2, EmiMIC3) has been evaluated by means of homogenous challenge animal experiments, and the results revealed that the MICs provided promising protection against *Eimeria* infection [37,43,44,45,46]. Similarly, ROMs are also important invasion associated antigens, they can cleave MICs in the transmembrane region, cut off the connection between MICs and the host cell ligand at the late stages of invasion, and finally, can cause the parasite to enter the cell completely [47,48]. Some reports on protozoan ROMs have studied the immune protection of ROMs. In *T. gondii*, Li and Zhang et al. determined the immune effect of pVAX–TgROM1, pVAX–TgROM4, and pVAX–TgROM5; they found that these TgROMs provided partial immune protection against *T. gondii* infection in mice [20,21]. In studies of the *E. tenella* rhomboid-like protein, Yang et al. constructed a recombinant fowlpox virus (rFPV) that could express the rhomboid gene of *E. tenella,* and Li et al. expressed its recombinant protein in *E. coli*. Both forms produced good immune protection against homologous challenge [18,19]. In this study, we found that EmROM5 provided moderate protective efficacy against *E. maxima* infection, which perhaps demonstrated that EmROM5 plays a certain role in the invasion of *E. maxima* from the other side. Nevertheless, this deduction requires further verification via specific experiments in the future.

Good immunogenicity is not only a necessary characteristic of a candidate antigen, but it is also a prerequisite for the vaccine to play an immune protective role. The immune response of chickens against coccidiosis is mainly mediated by cellular immunity involving T lymphocytes [49]. CD8^+^ T cells increase in number and play a major role in the secondary infection of parasites [50]; some studies have reported that CD8^+^ T cells come into direct contact with intestinal epithelial cells that have been invaded by parasites to destroy the infected cells [49,51]. CD4^+^ T cells increase significantly after primary infection and secrete a variety of cytokines that regulate cellular and humoral immunity [50]. Although the role of the humoral immune response in chicken infection with coccidia is controversial [49,50], some studies have shown that it has a certain relationship with immune protection. Parasite-reactive serum IgG is usually detected within 1 week after the oral infection of *Eimeria* oocysts [52]. The specific antibody can be transmitted to the filial generation through egg yolk and prevents *Eimeria* infection for a long time [53,54]. In this study, we determined the immunogenicity of EmROM5 in the form of a subunit vaccine and DNA vaccine by detecting the cellular immune response (T lymphocyte subpopulation, and cytokine level) and humoral immune response (specific IgG level). We found that in vaccinated chickens, immunization with EmROM5 increased the CD4^+^ and CD8^+^ T lymphocyte proportions in the spleens as well as the levels of the mRNA levels of six cytokines and the level of the serum-specific IgG. These results suggest that EmROM5 induced robust cellular and humoral immune responses in the immunized chickens, showing good immunogenicity.

It is worth mentioning that many cytokines play important and complex roles in anticoccidial immunity [55]. After *Eimeria* infection, the T lymphocytes in chickens can secrete a variety of cytokines, such as Th1-type (IFN-γ and IL-2), Th2-type (IL-4), and Th17-type (IL-17) cytokines; regulatory cytokines (TGF-β and IL-10); TNF, and so on [56,57]. Th1-type cytokines play major roles against *Eimeria* infection [32,57]. IFN-γ is a core cytokine that plays an anticoccidial role by mediating Th1 cell response, and it can inhibit the intracellular development of *Eimeria*. IL-2 can promote the growth and differentiation of many immune cells, such as T, B, NK cells [56,57]. In our results, the mRNA levels of both cytokines increased, indicating the strong activation of anticoccidial immunity in the immunized chickens. In addition, IL-4 can regulate humoral immunity and can promote B cell development and antibody production [58]. The mRNA level of IL-4 and the specific serum IgG level that we determined increased consistently, a finding that is consistent with what has been described in the literature. They both indicate that humoral immunity plays a certain role in resisting *Eimeria* infection. Moreover, the up-regulation of IL-17 and TNF after *Eimeria* infection can promote the production of pro-inflammatory response. They cooperate with other cytokines to play roles that are involved in both anti-infection and in the killing of invasive parasites [32,57,59,60]. Generally, the immune response of the body is a two-way regulation process. There are anti-inflammatory cytokines that are secreted by Treg cells. TGF-β is an inhibitory cytokine with immunomodulatory functions. It promotes the repair of the damaged intestinal epithelium and inhibits the proliferation of T and B cells [32,61]. IL-10 can inhibit the occurrence of host self-injury and can reduce the production of pro-inflammatory cytokines [62]. The production of these two inhibitory cytokines is also essential for immune protection against coccidiosis.

Lastly, although the DNA vaccine and subunit vaccine of EmROM5 only demonstrated moderate protective effects against *E. maxima*, there are additional measures that can be used to improve the protective efficacy of these vaccines. Some cytokines can be used as immune adjuvants to enhance the effects of the vaccines. The protective effect of the DNA vaccine can be strengthened by co-immunization with plasmids containing the cytokine genes of IL-8, IFN-γ, IL-15, or IL-1β [63]. Additionally, antigen and cytokine genes such as IL-2, IL-15, IFN-γ were combined into a plasmid for expression to enhance immune response [64,65]. In the case of the combined injection of Freund’s adjuvant and the ISA 71 VG adjuvant, the protective effect of the subunit vaccine was improved [66,67]. In addition, in order to improve the vaccine effect, we can also optimize the immunization procedure in terms of the dose, route of vaccination, vaccination age, and interval time, etc. [68]. Therefore, the application of EmROM5 as a candidate antigen in DNA vaccines and in subunit vaccines in broiler production requires further research in order to obtain the best immune effect.

## 5. Conclusions

The *E. maxima* ROM5 can significantly induce cellular and humoral immune responses and can provide moderate protection against *E. maxima* infection. All of the results demonstrated that EmROM5 is a promising candidate antigen for DNA vaccine and subunit vaccine development against clinical chicken coccidia infection.

## Figures and Tables

**Figure 1 vaccines-10-00032-f001:**
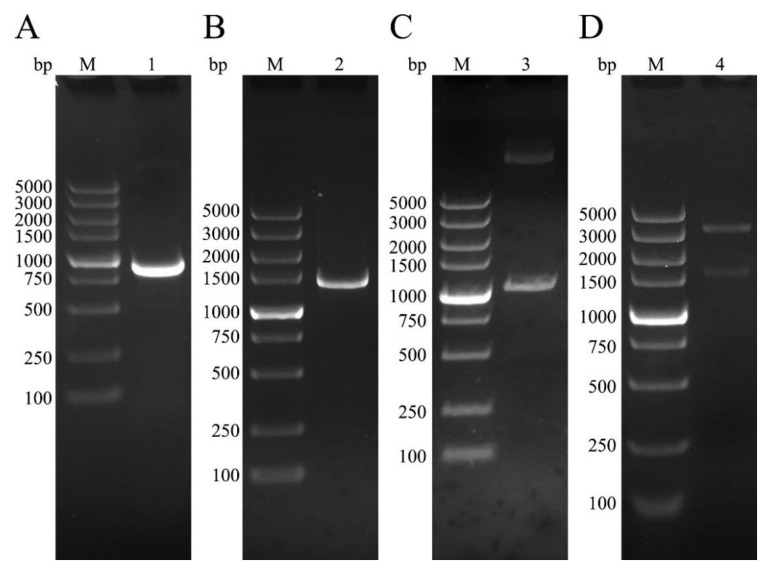
Gene cloning and vector construction. (**A**) RT-PCR amplification of ntmEmROM5. M, DNA molecular weight standard of DL5000. Lane 1, amplification products of ntmEmROM5 (963 bp). (**B**) RT-PCR amplification of EmROM5. M, DNA molecular weight standard of DL2000. Lane 2, amplification products of EmROM5 (1461 bp). (**C**) Enzyme digestion identification of pET-32a–ntmEmROM5. M, DNA molecular weight standard of DL2000. Lane 3, enzyme digestion identification of pET-32a–ntmEmROM5 (963 bp). (**D**) Enzyme digestion identification of pVAX–EmROM5. M, DNA molecular weight standard of DL5000. Lane 4, enzyme digestion identification of pVAX-EmROM5 (1461 bp).

**Figure 2 vaccines-10-00032-f002:**
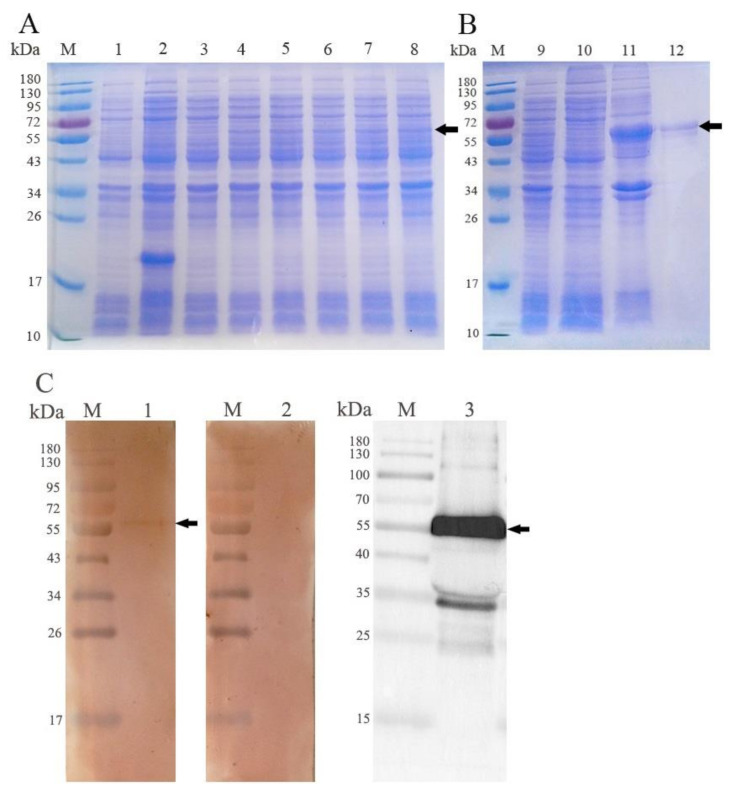
Expression of rEmROM5 and Western blot analysis. (**A**) Induction of rEmROM5 (53.64 kDa) by IPTG for 1–5 h. M, protein standard molecular weight. Lane 1, pET-32a-transfected bacteria before the induction of IPTG. Lane 2, pET-32a-transfected bacteria induced by IPTG for 5 h. Lane 3, pET-32a–ntmEmROM5-transfected bacteria before the induction of IPTG. Lanes 4–8, pET-32a–ntmEmROM5-transfected bacteria induced by IPTG for 1–5 h. (**B**) Purification of rEmROM5 (53.64 kDa). M, protein standard molecular weight. Lane 9, pET-32a–ntmEmROM5-transfected bacteria induced by IPTG for 5 h. Lane 10, cell lysate supernatant of pET-32a–ntmEmROM5 bacterial liquid after 5 h induction of IPTG. Lane 11, cell lysate sediment of pET-32a–ntmEmROM5 bacterial liquid after 5 h induction of IPTG. Lane 12, pET-32a–ntmEmROM5 recombinant protein (rEmROM5, 53.64 kDa) after purification. (**C**) WB analysis of rEmROM5 (53.64 kDa). M, protein standard molecular weight. Lane 1, recognition of rEmROM5 (53.64 kDa) by anti-*E. maxima* chicken serum. Lane 2, uninfected chicken serum control. Lane 3, recognition of rEmROM5 (53.64 kDa) by His-Tag Mouse Monoclonal antibody.

**Figure 3 vaccines-10-00032-f003:**
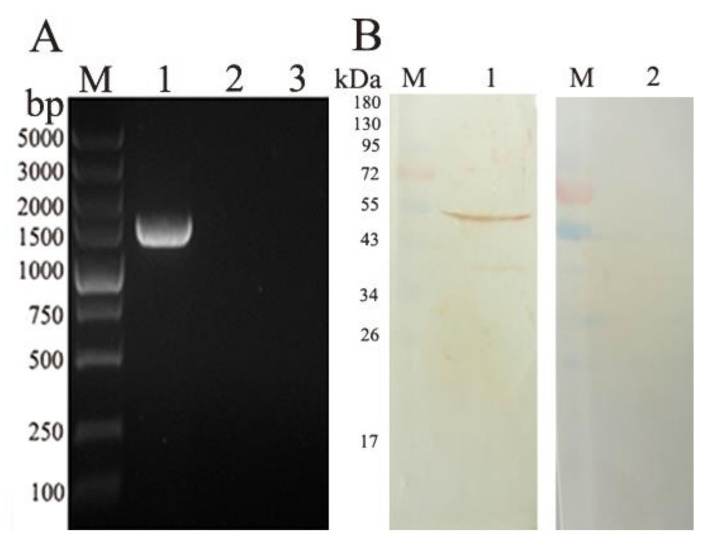
Transcription and expression of pVAX–EmROM5 in the injected muscles in chickens. (**A**) Transcription detection of pVAX–EmROM5 in the injected muscles by RT-PCR. M, DNA molecular weight standard of DL5000. Lane 1, PCR product of EmROM5 (1461 bp) from pVAX–EmROM5 injection site muscles. Lane 2, pVAX1.0 injection control. Lane 3, non-injected control. (**B**) Expression detection of EmROM5 in the injected muscles by WB. M, protein standard molecular weight. Lane 1, recognition of EmROM5 (53.57 kDa) in pVAX–EmROM5-injected muscles by anti-rEmROM5 rat serum. Lane 2, negative rat serum control.

**Figure 4 vaccines-10-00032-f004:**
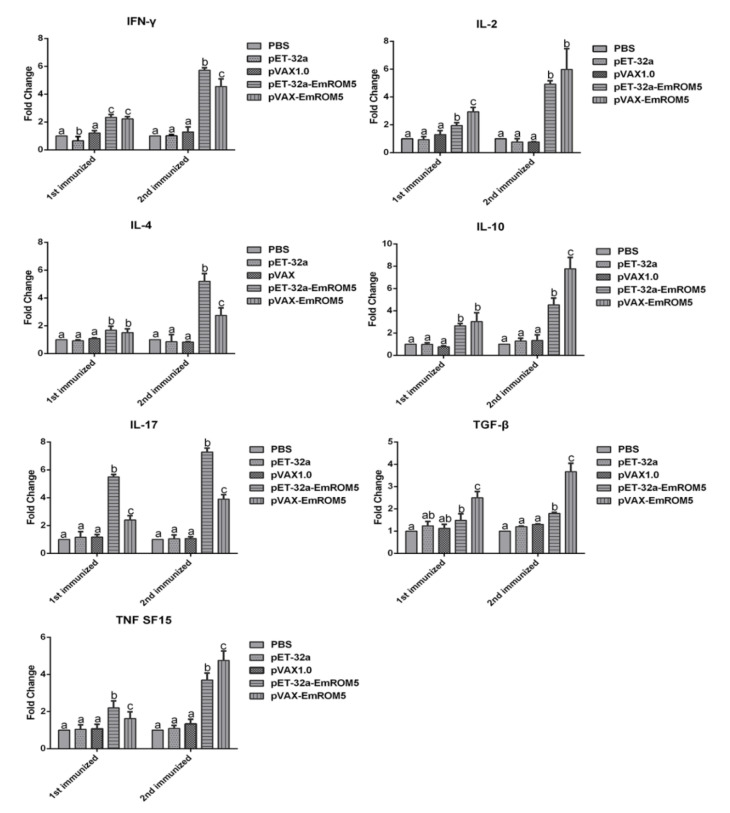
Changes in mRNA transcription of cytokines in splenic lymphocytes following the immunization of rEmROM5 and pVAX-EmROM5 (*n* = 5, value = mean ± SD). Significant difference (*p* < 0.05) between numbers with different letters, and no significant difference (*p* > 0.05) between numbers with the same letter.

**Figure 5 vaccines-10-00032-f005:**
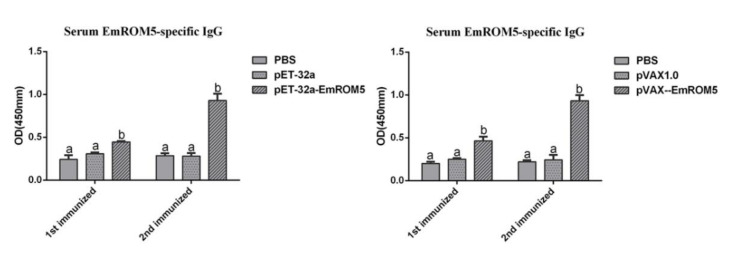
Serum EmROM5-specific IgG levels after immunization with rEmROM5 and pVAX–EmROM5 (*n* = 5, value = mean ± SD). Significant difference (*p* < 0.05) between numbers with different letters, and no significant difference (*p* > 0.05) between numbers with the same letter.

**Table 1 vaccines-10-00032-t001:** Specific primers of *E. maxima* ROM5.

Gene	Primer	Size (bp ^3^)
ntmEmROM5 ^1^	Forward: 5’-CCGGAATTCATGTCTTCCCCCATTG-3’	963
	Reverse: 5’-CCCTCGAGATGCAAAAAGGAGGCCCAAAAGAC-3’	
EmROM5 ^2^	Forward: 5’-CCGGAATTCATGTCTTCCCCCATTG-3’	1461
	Reverse: 5’-AAATATGCGGCCGCTCAAGTAAACTT-3’	

^1^ Non-transmembrane *E. maxima* ROM5; ^2^
*E. maxima* ROM5; ^3^ base pair.

**Table 2 vaccines-10-00032-t002:** Specific primer sequences of quantitative real-time PCR.

RNA Target	Primer Sequence	Accession No.	Amplification Efficiency (%)	Correlation Coefficient (r^2^)
GAPDH	GGTGGTGCTAAGCGTGTTAT	K01458	100.74%	0.9917
	ACCTCTGTCATCTCTCCACA			
IL-2	TAACTGGGACACTGCCATGA	AF000631	102.44%	0.9921
	GATGATAGAGATGCTCCATAAGCTG			
IL-4	ACCCAGGGCATCCAGAAG	AJ621735	99.09%	0.9936
	CAGTGCCGGCAAGAAGTT			
IL-10	GGAGCTGAGGGTGAAGTTTGA	AJ621614	99.19%	0.9923
	GAAGCGCAGCATCTCTGACA			
IL-17	ACCTTCCCATGTGCAGAAAT	EF570583	100.24%	0.994
	GAGAACTGCCTTGCCTAACA			
IFN-γ	AGCTGACGGTGGACCTATTATT	Y07922	103.07%	0.9868
	GGCTTTGCGCTGGATTC			
TGF-β	CGGGACGGATGAGAAGAAC	M31160	102.79%	0.9815
	CGGCCCACGTAGTAAATGAT			
TNF SF15	GCTTGGCCTTTACCAAGAAC	NM001024578	100.57%	0.993
	GGAAAGTGACCTGAGCATAGA			

**Table 3 vaccines-10-00032-t003:** Protective efficacy of rEmROM5 and pVAX-EmROM5 against challenge with *E. maxima* (*n* = 30, value = mean ± SD).

Trials	Groups	Average Body Weight Gain (g)	Relative Body Weight Gain (%)	Mean Lesion Score	Average OPG (×10^5^)	ACI
1	Unchallenged control	56.91 ± 10.24 ^a^	100 ^a^	0 ± 0 ^a^	0 ± 0 ^a^	200
	Challenged control	27.21 ± 8.52 ^c^	47.81 ^c^	2.84 ± 0.88 ^c^	2.25 ± 0.94 ^c^	79.41
	pET-32a tag protein control	29.46 ± 11.25 ^c^	51.77 ^c^	2.66 ± 0.93 ^c^	2.15 ± 0.97 ^c^	85.17
	rEmROM5	49.36 ± 11.35 ^b^	86.73 ^b^	1.46 ± 0.52 ^b^	0.56 ± 0.48 ^b^	171.13
2	Unchallenged control	79.32 ± 9.59 ^a^	100 ^a^	0 ± 0 ^a^	0 ± 0 ^a^	200
	Challenged control	39.28 ± 9.72 ^c^	49.53 ^c^	2.83 ± 0.72 ^c^	2.81 ± 0.13 ^c^	81.23
	pVAX1.0 control	38.19 ± 15.39 ^c^	48.15 ^c^	2.75 ± 0.62 ^c^	2.80 ± 0.16 ^c^	80.65
	pVAX–EmROM5	65.95 ± 4.96 ^b^	83.14 ^b^	1.25 ± 0.75 ^b^	0.67 ± 0.19 ^b^	169.64

^a–c^ Means in the same columns marked with the same letter indicates that the difference between treatments is not significant (*p* > 0.05). Means in the same columns marked with a different letter indicates a significant difference between treatments (*p* < 0.05).

**Table 4 vaccines-10-00032-t004:** Quantification of T lymphocyte subpopulations in spleen seven days after immunizations (*n* = 5, value = mean ± SD).

Marker	Groups	1st Immunization	2nd Immunization
CD4^+^/CD3^+^	PBS buffer	18.27 ± 0.21 ^a^	19.47 ± 3.21 ^a^
	pET-32a tag protein	20.90 ± 1.64 ^a^	23.28 ± 1.62 ^a^
	rEmROM5	27.75 ± 1.35 ^bc^	30.07 ± 0.57 ^b^
	PBS buffer	10.18 ± 0.87 ^a^	14.13 ± 1.50 ^a^
	pVAX1.0	11.50 ± 2.05 ^a^	16.45 ± 2.05 ^a^
	pVAX-EmROM5	22.57 ± 1.85 ^b^	27.51 ± 4.95 ^b^
CD8^+^/CD3^+^	PBS buffer	21.45 ± 0.72 ^a^	22.80 ± 5.20 ^a^
	pET-32a tag protein	22.37 ± 1.53 ^a^	25.93 ± 3.39 ^a^
	rEmROM5	35.20 ± 5.20 ^b^	43.59 ± 6.76 ^b^
	PBS buffer	13.00 ± 1.65 ^a^	13.78 ± 1.61 ^a^
	pVAX1.0	15.60 ± 1.13 ^b^	17.24 ± 0.42 ^a^
	pVAX-EmROM5	22.37 ± 0.17 ^c^	31.40 ± 4.48 ^b^

^a–c^ Means in the same columns marked with the same letter indicates that the difference between treatments is not significant (*p* > 0.05). Means in the same columns marked with a different letter indicates a significant difference between treatments (*p* < 0.05).

## Data Availability

The data presented in this study are available within the article and Appendix A.

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
