# Peer review of "Eimeria maxima* Rhomboid-like Protein 5 Provided Partial Protection against Homologous Challenge in Forms of Recombinant Protein and DNA Plasmid in Chickens"

_vaccines, 2021, doi:10.3390/vaccines10010032_

Round 1

Reviewer 1 Report

The authors had constructed subunit vaccine (rEmROM5) containing the non-transmembrane amino acid sequence of EmROM5 and DNA vaccine (pVAX-EmROM5) containing full length of EmROM5 gene against Eimeria maxima in chickens. These two vaccines induced both humoral and cellular immune responses including specific anti- EmROM5, CD4+ or CD8+ T cells, various cytokines. After animal trials, the moderate protection of these two vaccines against E. maxima was observed according to significantly higher body weight gain, and Anticoccidial index (ACI) or significantly lower lesion score and oocysts production (OPG) when compared to the challenge control groups. Author mentioned that the results indicated the EmROM5 could be used as a candidate antigen for DNA vaccines and subunit vaccines against avian coccidiosis.

Suggestions:

  • Line 88, Since the chickens were from commercial hatchery, did they have maternal antibody against Eimeria maxima?
  • Line 150, “China)at 37℃” changes to “China) at 37℃”.
  • Line 230, Is there any adjuvant used for subunit or DNA vaccine? The adjuvant is very important for preparing subunit vaccine. For DNA vaccine, naked plasmid may need higher doses (large amount and more times of injection) to induce proper immune responses.

  • Line 237, “The survival rate,” was mentioned here, but it was not presented in this paper for evaluating the protection efficacy.

  • Line 238-239, “oocyst output” and “Anticoccidial index (ACI)” may need to be briefly explained how to get these two data.

  • Line 241, “All data were analyzed by one-way analysis of variance (ANOVA)”, When comparing all the groups together, the results of ANOVA need to be tested with various methods (such as Duncan or Tukey test). Which test was used here?

  • In Figure 1,
    • The bands in all gels were not clear (might load too much amount of DNA) and better to rerun them.
    • The sizes of bands (for all 4 panels) need to be mentioned in the figure legend.
    • In Panel A, the size of band in Lane 1 shall be 963 bp. Why is the size of DNA in the Panel A larger than 1000 bp (Lane M).

  • In Figure 2,
    • All the important bands in the gels or blots might need to be pointed out by arrowheads.
    • The bands in Lane 1 and 2 of Panel C are not clear.
    • The KDa of bands (for all 3 panels) need to be mentioned in the figure legend.

  • In Figure 3, The problems occurred in Figure1 and 2 are also found here.

  • In Table 3 and Figure 4,
    • Since the immune responses induced by the subunit vaccine and DNA vaccine are not all based on the same mechanism, subunit vaccine induce more humoral immune response and DNA vaccine can also induce cellular immune responses. (If the adjuvant are used here, the immune responses might be redirected)
    • Both subunit vaccine and DNA vaccine show the same patterns of immune responses in Table 3 and Figure 4. The authors might need to explain them more.

  • In Figure 5, The antibody titer (indicated by the level of OD450 reading) shall be higher in the subunit vaccine than that in the DNA vaccine. However, the titers are similar in both subunit and DNA vaccine. Need to be discussed more.

Author Response

Thanks for your kind consideration of our manuscript. We have revised the entire manuscript thoroughly about all the issues mentioned in the reviewer' s comments. The following are responses to the specific issues raised by the reviewer. We hope that you will be satisfied with the revised version, if not, please give us another chance to revise or explain.

Point 1: Line 88, Since the chickens were from commercial hatchery, did they have maternal antibody against Eimeria maxima?

Response 1: The maternal antibody of chicks mainly comes from the hens whom were vaccinated with anticoccidial vaccines or infected by the wild coccida. The hens in the chicken farm where we bought chicks were not vaccinated with any coccidial vaccine. Therefore, there was no maternal antibody caused by vaccination of hens. Since the hens were raised in cages with good animal husbandry, the rate of natural infection with coccidia were very low, which meant that the probability of chicks with maternal antibodies was also very low. Even with low levels of maternal antibody, the maternal antibody almost disappeared 14 days after hatching. Therefore, the issue of maternal antibody has no essential impact on this trial.

Point 2: Line 150, “China)at 37℃” changes to “China) at 37℃”.

Response 2: We have revised it as required in line 157 of the revised manuscript (without track changes).

Point 3: Line 230, Is there any adjuvant used for subunit or DNA vaccine? The adjuvant is very important for preparing subunit vaccine. For DNA vaccine, naked plasmid may need higher doses (large amount and more times of injection) to induce proper immune responses.

Response 3: we did not use any adjuvant used for subunit or DNA vaccine. We have clarified this in line 238 of the revised manuscript (without track changes).

 In this study, although we did not use any adjuvant, moderated protection was achieved. In some researches on Eimeria recombinant vaccines, the birds vaccinated with Eimeria recombinant protein without adjuvants also showed effective efficacy (Crane et al., 1991, Ding et al., 2004, Zhang et al., 2012). For example, Crane et al(1991) demonstrated that a single dose of recombinant CheY-SO7' of E. tenella, without adjuvant, not only protected against severe coccidiosis induced by infection with E. tenella but also protected chicks challenged with the heterologous species Eimeria acervulina, E. maxima, and E. necatrix. Ding et al (2004) reported that vaccination with 100 or 500 μg of recombinant protein 3-1E without adjuvants resulted in significantly decreased oocyst shedding compared with that in nonvaccinated chickens. Zhang et al (2012) demonstrated that chickens immunized with recombinant EtMIC2 plus EtHSP70 without adjuvants showed increased body weight gains, decreased oocyst shedding, increased serum antibody responses, and high levels cytokine expression compared with control groups. Hence, for some antigens, they might induce effective efficacy against Eimeria challenge even administrated without adjuvant. Although adjuvants do improve the efficacy of vaccines, however, the traditional adjuvants might cause side effects and safety problems (Aucouturier et al., 2001; Cox and Coulter, 1997; O'Hagan et al., 2001). Hence, when these vaccines are developed for field use, whether the adjuvant is included and what type of adjuvant is used need further researches in the future. We discussed this issue in line 463-468 of the revised manuscript.

The dose of 100 μg was used in many researches on vaccination of naked DNA plasmid. Hence, we also used this vaccination dose and got moderate protective efficacy against E. maxima. Interestingly, more plasmid correlated with stronger response, but not necessarily linearly (Barry and Johnston, 1997; Leitner et al., 1997). Among chickens immunized intramuscularly with various doses of pMP13 expression vector ranging from 5 μg to 100 μg, both 5 μg and 50 μg DNA group presented more effective in reducing the oocyst production after challenge with 5 ×103 E. acervulina. However, the highest dose group of 100 μg was the worst one (Song et al., 2001). In our previous research, the ACI of 25 μg group was the highest followed by 50 μg and 100 μg. However, 200 μg was the lowest (Song et al., 2009). Caputo et al. (2003) reported that the strongest immune response against Tat was elicited by the dose of 10 μg, while immunization with 30 μg of naked tat was less effective and the reason was unclear. In our future research, we are considering to optimize the immunization procedure, including larger doses and more times of injection to improve the protective efficacy of the vaccines.

References:

Aucouturier, J., Dupuis, L., Ganne, V., 2001. Adjuvants designed for veterinary and human vaccines. Vaccine, 19(17-19), 2666-2672.

Cox, J.C., Coulter, A.R., 1997. Adjuvants-a classification and review of their modes of action. Vaccine, 15(3), 248-256.

O'Hagan, D.T., MacKichan, M.L., Singh, M., 2001. Recent developments in adjuvants for vaccines against infectious diseases. Biomolecular Engineering, 18(3):69-85.

Crane, M.S., Goggin, B., Pellegrino, R.M., Ravino, O.J., Lange, C., Karkhanis, Y.D., Kirk, K.E., Chakraborty, P.R., 1991. Cross-protection against four species of chicken coccidia with a single recombinant antigen. Infect Immun, 59(4), 1271-1277.

Ding, X., Lillehoj, H.S., Quiroz, M.A., Bevensee, E., Lillehoj, E.P., 2004. Protective immunity against Eimeria acervulina following in ovo immunization with a recombinant subunit vaccine and cytokine genes. Infect Immun, 72(12), 6939-6944.

Zhang, L., Ma, L., Liu, R., Zhang, Y., Zhang, S., Hu, C., Song, M., Cai, J., Wang, M., 2012. Eimeria tenella heat shock protein 70 enhances protection of recombinant microneme protein MIC2 subunit antigen vaccination against E. tenella challenge. Vet Parasitol, 188(3-4), 239-246.

Barry, M.A., Johnston, S.A., 1997. Biological features of genetic immunization. Vaccine 15, 788–791.

Leitner, W.W., Seguin, M.C., Ballou, W.R., Seitz, J.P., Schultz, A.M., Sheehy, M.J., Lyon, J.A., 1997. Immune responses induced by intramuscular or gene gun injection of protective deoxyribonucleic acid vaccines that express the circumsporozoite protein from Plasmodium berghei malaria parasites. J. Immunol. 159, 6112–6119.

Song, K.D., Lillehoj, H.S., Choi, K.D., Yun, C.H., Parcells, M.S., Huynh, J.T., Han, J.Y., 2001. A DNA vaccine encoding a conserved Eimeria protein induces protective immunity against live Eimeria acervulina. Vaccine 19, 243–252.

Song, X., Xu, L., Yan, R., Huang, X., Shah, M. A., & Li, X. (2009). The optimal immunization procedure of DNA vaccine pcDNA-TA4-IL-2 of Eimeria tenella and its cross-immunity to Eimeria necatrix and Eimeria acervulina. Veterinary parasitology159(1), 30–36.

Caputo, A., Gavioli, R., Altavilla, G., Brocca-Cofano, E., Boarini, C., Betti, M., Castaldello, A., Lorenzini, F., Micheletti, F., Cafaro, A., Sparnacci, K., Laus, M., Tondelli, L., Ensoli, B., 2003. Immunization with low doses of HIV-1 tat DNA delivered by novel cationic block copolymers induces CTL responses against Tat. Vaccine 21, 1103–1111.

Point 4: Line 237, “The survival rate,” was mentioned here, but it was not presented in this paper for evaluating the protection efficacy.

Response: The survival rate was used to calculate the Anticoccidial index (ACI), which has been clarified in line 247-253 of the revised manuscript (without track changes).

Point 5: Line 238-239, “oocyst output” and “Anticoccidial index (ACI)”μ may need to be briefly explained how to get these two data.

Response: Thanks for your constructive suggestion. We have explained how to get these two data in line 250-253 of the revised manuscript(without track changes). 

Point 6: Line 241, “All data were analyzed by one-way analysis of variance (ANOVA)”, When comparing all the groups together, the results of ANOVA need to be tested with various methods (such as Duncan or Tukey test). Which test was used here?

 Response: Thanks for your constructive suggestion. We have clarified this in line 255-259 of the revised manuscript (without track changes).

Point 7: In Figure 1,

The bands in all gels were not clear (might load too much amount of DNA) and better to rerun them.

The sizes of bands (for all 4 panels) need to be mentioned in the figure legend.

In Panel A, the size of band in Lane 1 shall be 963 bp. Why is the size of DNA in the Panel A larger than 1000 bp (Lane M).

Response: Thanks for your constructive suggestion. We have reruned all the gels and replaced all the panels in Figure 1. Now, the size of DNA in the Panel A is correct.

Point 8: In Figure 2,

All the important bands in the gels or blots might need to be pointed out by arrowheads.

The bands in Lane 1 and 2 of Panel C are not clear.

The KDa of bands (for all 3 panels) need to be mentioned in the figure legend.

In Figure 3, The problems occurred in Figure1 and 2 are also found here.

Response: Thanks for your constructive suggestion. The target bands in figure 2 and figure 3 have been pointed out by arrowheads. The KDa of bands in figure 2 and figure 3 have been mentioned in the figure legends.

Point 10:

In Table 3 and Figure 4,

Since the immune responses induced by the subunit vaccine and DNA vaccine are not all based on the same mechanism, subunit vaccine induce more humoral immune response and DNA vaccine can also induce cellular immune responses. (If the adjuvant are used here, the immune responses might be redirected)

Both subunit vaccine and DNA vaccine show the same patterns of immune responses in Table 3 and Figure 4. The authors might need to explain them more. 

In Figure 5, The antibody titer (indicated by the level of OD450 reading) shall be higher in the subunit vaccine than that in the DNA vaccine. However, the titers are similar in both subunit and DNA vaccine. Need to be discussed more.

Response: Thanks for your constructive suggestion.  The intensity and type of immune response induced by vaccine may depend on many factors, such as immunization route, dose, times, form of the antigen and etc. In this study, we did not use any adjuvant for the subunit vaccine and DNA vaccine, and the immunization dose, times and route were not optimized. Hence, the intensity of immune response induced by vaccine may not be the most appropriate, which could explain why the similar patterns of immune responses in the subunit vaccine and DNA vaccine. In other words, if we have used the proper adjuvant, optimized immunization dose, route and times, the patterns of immune responses induced by the subunit vaccine and DNA vaccine could show obvious difference.

Reviewer 2 Report

The abstract needs to be rewritten incorporating the main results obtained.

In session 2.4 some important steps of protein purification are missing, complete or refer a technique followed from literature. Is also missing the production and purification process of the recombinant plasmid used for vaccination.

When referred to centrifugation rate please present instead of “r/min” present in “g”, in all manuscript.

The vaccination process is not clear which was the adjuvant used for recombinant protein and plasmid vaccine based.

Concerning statistical analysis of data: First the data for statistical analysis must be evaluated if they are normal distribution thus a parametric analysis can be used otherwise a non-parametric statistical analysis must be used.

Results:

If the recombinant protein has a His tag? why it was not used an anti-His antibody to demonstrate that His tag protein was the purified on in a western blot?

The recombinant protein was produced in pET-32a tag protein (Trx•Tag™ thioredoxin protein) if so then the His-tag column is not appropriate to purify the recombinant protein? Please clarify all these aspects that are not clear.

Table 3 the data presented are very different when two groups with PBS are compaired. The n is missing in the table to better data discussion. The data presented are mean of many results (n=?, mean±SD).

In data presented in session 3.5 is not clear how data were compared.

In figure 4 and 5 in the legend is missing the meaning of the letters under each data in the graphics. Also, must be add to the legend the (n=?, mean±SD).

Table 4 must be add to the legend the (n=?, mean±SD).

Author Response

Thanks for your kind consideration of our manuscript. We have revised the entire manuscript thoroughly about all the issues mentioned in the reviewer' s comments. The following are responses to the specific issues raised by the reviewer. We hope that you will be satisfied with the revised version, if not, please give us another chance to revise or explain.

Comments and Suggestions for Authors

Point 1: The abstract needs to be rewritten incorporating the main results obtained.

Response: Thanks for your constructive suggestion. We have made revisions on the abstract.

Point 2: In session 2.4 some important steps of protein purification are missing, complete or refer a technique followed from literature. Is also missing the production and purification process of the recombinant plasmid used for vaccination.

Response: Thanks for your constructive suggestion. We have added the missing production and purification information of the recombinant plasmid and protein in line 122-125 and line129-130 respectively, in the revised manuscript (without track changes).

Point 3: When referred to centrifugation rate please present instead of “r/min” present in “g”, in all manuscript.

Response: Thanks for your constructive suggestion. We have modified these.

Point 4: The vaccination process is not clear which was the adjuvant used for recombinant protein and plasmid vaccine based.

Response: Thanks for your constructive suggestion. We have clarified these in line 238 of the revised manuscript.

Point 5: Concerning statistical analysis of data: First the data for statistical analysis must be evaluated if they are normal distribution thus a parametric analysis can be used otherwise a non-parametric statistical analysis must be used.

Response: Thanks for your constructive suggestion. We have clarified these in section 2.9 of the revised manuscript.

 Results:

Point 6: If the recombinant protein has a His tag? why it was not used an anti-His antibody to demonstrate that His tag protein was the purified on in a western blot?

Response: Thanks for your constructive suggestion. We have carried out a western blot assay using an anti-His antibody as the primary antibody to demonstrate that the recombinant protein with His tag protein. The results was presented in Figure 2C (lane 3).

Point 7: The recombinant protein was produced in pET-32a tag protein (Trx•Tag™ thioredoxin protein) if so then the His-tag column is not appropriate to purify the recombinant protein? Please clarify all these aspects that are not clear.

Response: According to the manufacture’s instruction, the pET-32 vector contains His Tag® and S•Tag™ sequences for detection and purification. In this study, the recombinant protein was expressed through pET-32 vector, producing the fusion protein with His tag. Therefore, HisTrap TM FF Column is appropriate to purify the recombinant protein. In fact, Histrap TM FF column is often used to purify recombinant protein expressed by pET-32a in many researches.

Point 8: Table 3 the data presented are very different when two groups with PBS are compaired. The n is missing in the table to better data discussion. The data presented are mean of many results (n=?, mean±SD).

Response: Thanks for your constructive suggestion. We have added the missing information in legend of Table 3. In this study, we carried out two experiments to determine the immune response induced by rEmROM5 and pVAX-EmROM5 separately. That is too say, two groups with PBS belonged to 2 separate experiments, which might be the reason why the data presented in the two groups with PBS are very different.

Point 9: In figure 4 and 5 in the legend is missing the meaning of the letters under each data in the graphics. Also, must be add to the legend the (n=?, mean±SD).

Table 4 must be add to the legend the (n=?, mean±SD).

Response: Thanks for your constructive suggestion. We have added the missing information in in the legend of in Figure 4, Figure 5 and Table 3.

Round 2

Reviewer 2 Report

Authors have answer to all questions and considered the suggestions, thus the paper can be considered for publication.